Head capsule stacking by caterpillars: morphology complements behaviour to provide a novel defence

Low Petah A. petah.low@sydney.edu.au
McArthur Clare
Hochuli Dieter F.
School of Biological Sciences, The University of Sydney , Sydney, New South Wales , Australia
Barrett Louise
Electronic publication date: 2016 Feb 29
Publication date: 2016
Volume: 4
Electronic Location ID: e1714
Received 2015 Dec 5; Accepted 2016 Jan 29
Copyright: ©2016 Low et al.
Copyright year: 2016
Copyright holder: Low et al.
License: This is an open access article distributed under the terms of the Creative Commons Attribution License, which permits unrestricted use, distribution, reproduction and adaptation in any medium and for any purpose provided that it is properly attributed. For attribution, the original author(s), title, publication source (PeerJ) and either DOI or URL of the article must be cited.
License URL: https://creativecommons.org/licenses/by/4.0/

Keywords: Defence, Lepidoptera, Parasitoid, Uraba lugens, Survival, Predator

Funding: Australasian Society for the Study of Animal Behaviour This research was supported by a Student Research Grant to PAL from the Australasian Society for the Study of Animal Behaviour. The funders had no role in study design, data collection and analysis, decision to publish, or preparation of the manuscript.

==============================
Herbivores employ a variety of chemical, behavioural and morphological defences to reduce mortality from natural enemies. In some caterpillars the head capsules of successive instars are retained and stacked on top of each other and it has been suggested that this could serve as a defence against natural enemies. We tested this hypothesis by comparing the survival of groups of the gumleaf skeletoniser Uraba lugens Walker caterpillars, allocated to one of three treatments: “−HC,” where stacked head capsules were removed from all individuals, “+HC,” where the caterpillars retained their stacked head capsules, and “mixed,” where only half of the caterpillars in a group had their stacked head capsules removed. We found no difference in predation rate between the three treatments, but within the mixed treatment, caterpillars with head capsules were more than twice as likely to survive. During predator choice trials, conducted to observe how head capsule stacking acts as a defence, the predatory pentatomid bug attacked the −HC caterpillar in four out of six trials. The two attacks on +HC caterpillars took over 10 times longer because the bug would poke its rostrum through the head capsule stack, while the caterpillar used its head capsule stack to deflect the bug’s rostrum. Our results support the hypothesis that the retention of moulted head capsules by U. lugens provides some protection against their natural enemies and suggest that this is because stacked head capsules can function as a false target for natural enemies as well as a weapon to fend off attackers. This represents the first demonstration of a defensive function.

Introduction

Natural enemies strongly influence the survival and fitness of insect herbivores and consequently are thought to have played a significant role in their evolution (Price et al., 2011; Schoonhoven, Van Loon & Dicke, 2005). Caterpillars in particular are heavily preyed upon by a huge array of true predators, both vertebrate and invertebrate, and are host to a diversity of parasitic arthropods (Scoble, 1992). In response, caterpillars have evolved a variety of ways to reduce mortality from natural enemies, including chemical, behavioural and morphological defenses (Stamp & Casey, 1993). For instance, many caterpillars possess defensive glands, emit offensive odors or sequester chemicals from their host plants to make themselves toxic or unpalatable (Bowers, 1993; Bowers, 2003). A range of behaviours can also form part of their defensive repertoire. Examples include resting on the underside of leaves, the building and use of refuges (Tvardikova & Novotny, 2012), removing evidence of their presence by throwing frass (Weiss, 2003) or clipping damaged leaves (Edwards & Wanjura, 1989; Heinrich & Collins, 1983; Weinstein, 1990), feeding gregariously (McClure & Despland, 2011; Reader & Hochuli, 2003), reducing activity (Thaler & Griffin, 2008), emitting startle or warning sounds such as clicking or whistling (Brown, Boettner & Yack, 2007; Bura et al., 2011), regurgitating gut contents (Grant, 2006), and thrashing, rearing or dropping from the plant (Allen, 1990a; Castellanos et al., 2011; Low, McArthur & Hochuli, 2014). Morphological defenses are also pervasive and include modifications for crypsis or camouflage such as shape disruption, color matching and counter-shading (Hossie & Sherratt, 2012; Rowland et al., 2008; Stamp & Wilkens, 1993), aposematic coloration (Bernays & Montllor, 1989), as well as the presence of protective hairs or spines (Murphy et al., 2010).

Understanding how prey defences interact with and influence the foraging of natural enemies is key to understanding the process of predation, and ultimately to understanding the crucial role that natural enemies play in regulating populations of insect herbivores and preventing the depletion of plant resources. Defences are thought to increase prey survival in the presence of natural enemies. In the past, the study of defence has largely been anecdotal and based on intuitive and subjective interpretations (Malcolm, 1992; Scoble, 1992), and consequently, in many cases the evidence for a defensive strategy is purely circumstantial (Malcolm, 1992; Scoble, 1992). However, more recently there have been a growing number of rigorous and objective experimental studies that actually demonstrate a defensive function (e.g., Castellanos et al., 2011).

In some nolid caterpillars (Lepidoptera: Nolidae), the head capsules of successive instars are retained and stacked on top of each other above the head, a peculiar behaviour or developmental phenomenon which has been recorded in a number of species throughout the old world including Mimerastria mandshuriana in Japan, Roeselia togatulalis and R. nitida in Europe, Rhynchopala argentalis in India and the Australian native Uraba lugens (McFarland, 1980; Fig. 1). It has been suggested that the stack of moulted head capsules could act as a defence, for instance by providing a false target for predators (McFarland, 1980; Scoble, 1992). In addition to this “decoy mechanism” (which has also been referred to as a divertive or deflective effect), the head capsules could make the caterpillar appear larger or more formidable to a potential predator (“illusion mechanism”) or be used in combination with thrashing and rearing behaviours to fend off enemies, including those attacking from behind (“lance mechanism”). Indeed, U. lugens often rear and thrash in response to simulated attack (Low, McArthur & Hochuli, 2014) and actual attack (Allen, 1990a), behaviours which are known to reduce the likelihood of being parasitized (Allen, 1990a). However, these hypotheses have not been tested and therefore the purpose (if any) of retaining moulted head capsules remains a mystery.

The gumleaf skeletoniser Uraba lugens has a wide distribution throughout Australia (Campbell, 1962), feeding predominantly on myrtaceous tree species, including a variety of Eucalyptus and Angophora species (Berndt & Allen, 2010). During the first four instars, feeding occurs gregariously with larvae skeletonizing leaves, while older larvae begin to disperse, feeding individually and consuming almost the entire leaf (Cobbinah, 1978). Larvae go through a variable number of instars (usually 8–14) and retain their moulted head capsules from about the fifth instar (Berndt & Allen, 2010; Cobbinah, 1978) (Fig. 1). Mature larvae grow to about 20–25 mm in length. They are well defended with urticating hairs which are thought to protect them from predation by birds (Allen, 1990b). However, they are heavily attacked by a wide range of parasitic wasps and flies (Allen, 1990b; Berndt & Allen, 2010; Farr, 2002) as well as predatory bugs, jumping spiders and lacewings (Berndt & Allen, 2010).

Figure 1 Uraba lugens caterpillars (approximately sixth instar) with moulted head capsules stacked above their heads.

Photo: P. Low.

We manipulated head capsule stacks on larvae to investigate the putative defensive function of head capsule stacking, testing the hypothesis that the retention of moulted head capsules by U. lugens caterpillars decreases rates of predation and parasitism in the field. We also aimed to investigate the mechanisms by which head capsule stacking could serve as a defence, through observations made during predator choice trials.

Materials and Methods

Field observations

We surveyed thirty natural groups of Uraba lugens larvae (∼5th–6th instar) at our field site in Sydney Harbour National Park, New South Wales (151°15′30″E, 33°49′45″S) in the summer of 2013–2014. We recorded the number of individuals in each group and the number of head capsules on each caterpillar. A caterpillar was considered to be part of a group if it was touching or within a body length of another (Reader & Hochuli, 2003). We used these observations to calculate the average number of caterpillars per group and assess head capsule distributions.

Influence of stacked head capsules on rates of predation and parasitism

We collected recently hatched U. lugens from our field site during the summer of 2013–2014 and took them back to the laboratory for rearing to minimize the chance of caterpillars being parasitized before the experiment. The caterpillars were housed in plastic containers and kept in a constant temperature room, maintained at 23 ± 2 °C with approximately 90% humidity. They were regularly provided with fresh foliage from Angophora floribunda (Sm.) Sweet, one of their host plant species. When caterpillars reached approximately their fifth or sixth instar (larvae about 10 mm long, with 2–3 head capsules), they were returned to the field. We set the caterpillars on Angophora and Eucalyptus trees in groups of ten individuals, with each group placed on a separate tree. This group size was chosen based on field observations of group sizes for caterpillars of this stage. The groups were allocated to one of three treatments: “−HC,” where stacked head capsules were removed from all ten individuals, “+HC,” where the caterpillars retained their stacked head capsules, and “mixed”, where five of the caterpillars had their stacked head capsules removed and five retained their head capsule stack. The “mixed” treatment was included to investigate any social dimension to the defence, in other words, whether the benefit of head capsule stacks as a defence is dependent on “easier,” less defended alternatives being available. The stacked head capsules were easily removed by gently lifting them from off the top of the caterpillar’s head using soft forceps, while holding the caterpillar down with a soft paint brush. Caterpillars that retained their stacked head capsules experienced a similar level of disturbance. We set twenty groups per treatment type and the host plant species used were evenly represented among treatments. Further, we blocked replicates from each treatment in space to minimise any effect of spatial variation in risk. Survival was monitored after four and eight days. We were confident that caterpillars would be recovered if present because they leave obvious signs of their presence in the form of feeding damage and moulted skins. On day four, we removed any new head capsules from the relevant caterpillars. All caterpillars remaining at the end of the eight days were collected and taken back to the laboratory. Here they were reared through to adults (∼16 weeks) to allow assessment of rates of parasitism. Caterpillars from each replicate were housed separately in plastic containers in the CT room and fed A. floribunda foliage.

We compared the survival of caterpillars from the different treatments using Cox proportional hazards analysis. We performed two analyses, one comparing survival between the +HC and −HC group treatments, and a second comparing survival of +HC and −HC caterpillars within the mixed treatment. To control for the lack of independence of caterpillars within a group, we clustered the caterpillars by group in our analyses. Additionally, we stratified the analyses by host plant to account for any differences in hazard rate between tree species, and stratified the first analysis by replicate to account for the blocking of replicates. Statistical analyses were performed using R version 3.2.3 (R Development Core Team, 2015), and we tested the assumption of proportionality using the cox.zph function.

Field work was conducted under National Parks and Wildlife Services N.S.W. Scientific Licence number SL100838.

Observation of predation events

To investigate possible mechanisms by which head capsule stacks serve as a defence, we conducted predator choice trials. We used direct capture and beat sampling to collect potential arthropod predators from the Eucalyptus and Angophora trees at our field site. We then used the collected predators, which included various spiders (four Sparassidae, four Clubionidae, two Thomisidae and two Salticidae) and one pentatomid bug (Cermatulus nasalis Woodward, Pentatomidae), in predator choice trials. For the choice trials, we placed two caterpillars, in approximately their 6th instar and matched for size, into a Petri dish (9 cm diameter) with a single eucalypt leaf lying flat on the base. One of the caterpillars had its stack of moulted head capsules removed using soft forceps (−HC), while the other retained its moulted head capsules but received a similar amount of physical disturbance (+HC). A single predator was then introduced and given the choice of attacking either caterpillar. For spiders, we ran multiple trials simultaneously, watching them for the first half hour and then filming with a video camera for the rest of the 24 h. We tested the pentatomid bug six times and these trials were also watched and filmed. To increase their motivation for feeding, the spiders were starved for six days prior to a trial, while the pentatomid bug was starved for at least two days between successive trials. We recorded which caterpillar was attacked first (−HC or +HC), where the caterpillar was attacked (head, middle, abdomen), the caterpillar’s responses to attack and whether the attack was successful (i.e., prey killed). We also calculated attack duration, which for the pentatomid bug was defined as the time between when the bug extended its rostrum and when the rostrum was successfully inserted into the caterpillar.

Results

Field observations

Uraba lugens larvae occurred in groups of varying size (1–47 individuals), though most commonly, the caterpillars were found in small groups (Fig. 2), with an average of 10.4 ± 1.8 SE individuals and a median of 8 individuals (interquartile range = 5.75, n = 30 groups). Most groups were dominated by caterpillars with two or three stacked head capsules (Fig. 2). Although the individuals within a single group often had the same number of head capsules, eight out of the thirty groups contained caterpillars with differing numbers of head capsules.

Figure 2 The frequency of different group sizes for 30 groups of fifth–sixth instar Uraba lugens caterpillars observed in the field.

Shading indicates the dominant number of stacked head capsules possessed by caterpillars in each group.

Influence of stacked head capsules on rates of predation

Overall there was a 16% survival rate, with only 96 of the 600 caterpillars placed in the field surviving the eight days. There was no statistically significant difference in survival between the caterpillars in the +HC and −HC treatment groups (z = 3.96, df = 1, P = 0.05; Fig. 3A). However within the mixed treatment, caterpillars with head capsule stacks were more than twice as likely to survive as those without (z = − 2.83, df = 1, P = 0.005; Fig. 3B). In many instances where no caterpillars remained after eight days, feeding damage and moulted skins on the leaves indicated that the caterpillars had settled and established themselves, and in more than 13% of the replicates there was direct evidence of predation (caterpillar remains, Fig. S1).

Figure 3 Survival of caterpillars (A) from three treatments groups, +HC (head capsule stacks intact), −HC (head capsule stacks removed), and mixed (head capsule stack removed from half of the group), and (B) within the mixed treatment.

For each treatment, 200 caterpillars were set out in 20 groups consisting of 10 larvae.

Influence of stacked head capsules on rates of parasitism

Two types of parasitoid had attacked the caterpillars, a species of wasp from the family Braconidae which emerged from the larval stage and a species of fly from the family Tachinidae which emerged once the caterpillars had pupated. Of the 46 −HC caterpillars remaining at the end of the field experiment, seven of these were parasitized (15% parasitism rate), three by wasps and four by flies. In contrast, only two of the 50 surviving +HC caterpillars were parasitized (4% parasitism rate), both by flies.

Observation of predation events

None of the spiders attacked the caterpillars. The pentatomid bug, however, readily attacked them, attacking the −HC caterpillar in four out of the six choice trials. In all six trials, the bug attacked the head end of the caterpillar and eventually inserted its rostrum near the caterpillar’s head. When attacked, +HC and −HC caterpillars showed similar behavioural responses, including thrashing, rearing their head, curling their body, regurgitating and walking away. Attacks on +HC caterpillars took much longer (≥127s) than attacks on −HC caterpillars (≤14s). During attacks on +HC caterpillars, the bug poked its rostrum into the head capsule stack several times (Figs. 4A–4C; see also Video S1, e.g., 1:08, 1:15) and the caterpillars used their head capsule stack to fend off the bug and deflect its rostrum (Figs. 4D–4F; Video S1, e.g., 1:38, 1:54, 2:15, 2:37), all of which contributed to prolonging the attack duration. Although varying in duration, all attacks were eventually successful.

Figure 4 Screen captures from a video of an attack by the pentatomid bug on a Uraba lugens caterpillar, showing how the caterpillar uses its head capsule stack to defend itself; head capsule stack (A–C) serving as a false target or decoy, (D–F) being used to deflect the bug’s rostrum.

For the full video, see Video S1.

Discussion

Our results support the hypothesis that the retention of moulted head capsules by Uraba lugens provides some protection against their natural enemies and suggest that this is at least partially because stacked head capsules can function as a false target for natural enemies as well as a weapon to fend off some attackers.

Our field experiment confirmed predation as an important source of mortality for U. lugens larvae, since the mortality rate was 84% over just 8 days (∼¼ of their larval duration). Defences are unlikely to be effective against the full suite of natural enemies in natural systems, and so such high mortality levels are not surprising or unusual. However, greater survival of caterpillars with head capsule stacks within our ‘mixed’ treatment groups and reduced levels of parasitism among surviving +HC caterpillars, together provide some evidence that under certain conditions head capsule stacking can be an effective defence, more than doubling a caterpillar’s chance of survival. That a difference in predation rate between caterpillars with and without their head capsule stack was only found in the ‘mixed’ treatment groups, suggests a social dimension to the defence; a caterpillar’s vulnerability is influenced by the traits (i.e., the presence or number of stacked head capsules) of other members of the group. Importantly, the observational component of our study confirmed that under natural conditions caterpillars do occur in groups where individuals have varying numbers of stacked head capsules, and therefore that it is realistic that predators may have the opportunity to select between prey differing in their defence. The observed differences in head capsule numbers could result from the loss of head capsules or differences in the speed of development and timing of moulting. Although our study only compared the presence and absence of head capsules, it will also be important to consider how the number of stacked head capsules might influence the effectiveness of head capsule stacks as a defence.

Our field results also raise the possibility that predator type or the level of predation pressure may be important in influencing the effectiveness of head capsule stacking as a defence. There were some groups where survival was high and others where survival was poor, irrespective of whether the caterpillars had stacked head capsules or not. This could suggest that, while providing some level of protection, head capsules may not be equally effective against all predators and are not sufficient to prevent predation by highly motivated predators. It is also possible that when predation pressure is very high or very low, head capsules may not provide an advantage. Rather, head capsule stacking may be most effective as a defence under intermediate levels of predation pressure. Consequently we propose that small-scale spatial variation in predation pressure may have contributed to the variable influence of head capsule stacking on caterpillar survival seen in our study. Indeed, in a similar study testing the effectiveness of the fecal shields of tortoise beetle larvae as a defence against their predators, shields were found to be least effective under low prey density, likely because when prey availability is low relative to predator abundance (i.e., predation pressure is relatively high), hungry predators are less deterred by prey defences and more persistent in their attacks (Olmstead & Denno, 1993).

Field experiments such as ours, comparing the survival of defended and undefended prey, are rare but important because they allow examination in a natural setting of the effectiveness of a defence against an entire complex of natural enemies. An assumption, though, is that disappearance equates to predation. While it is possible that some of the disappearances of the larvae could have alternative explanations, such as dropping off the leaf or dispersal, we are confident that disappearance was largely the result of predation. Uraba lugens caterpillars leave obvious signs of their presence, in the form of feeding damage and moulted skins, and for many groups where no larvae were recovered there was evidence that they had settled and become established on the branches where they were set, making it unlikely that they just fell off the plant or dispersed from the area. In other cases there was more direct evidence of predation, most likely from spiders, which included drained bodies wrapped in silk and masticated remains containing caterpillar hairs and in some cases intact head capsule stacks (Fig. S1). Further, there is no reason to expect that there would be a systematic bias in rate of dispersal between our treatments, suggesting that any difference can reasonably be attributed to predation.

The lack of predation by the various spiders during the predator choice trials was surprising given the evidence of spider predation in the field experiment. This was possibly because the spiders responsible for attacking the caterpillars in the field were not among those collected. Alternatively, the spiders may have needed more than 24 h to set up webbing or establish hunting areas. The pentatomid bug, however, readily attacked the caterpillars. Although we lacked replicate bugs, the choice trials using the one bug showed that head capsule stacks can serve a defensive function. Additionally the trials did not provide evidence against any of the hypothesized mechanisms, but rather suggest that a combination of them could be operating. The tendency for the bug to attack the −HC individual more often than the +HC individual is consistent with the ‘illusion mechanism,’ assuming the bug uses sight to select prey and that previous learning did not play a part. For instance, it is possible that the bug had learnt that prey with head capsules take longer to subjugate or are less likely to result in successful predation, and are therefore less profitable. Indeed there is a growing body of work showing that predatory insects are capable of associative learning (e.g., Guillette, Hollis & Markarian, 2009). The substantially longer duration of attacks for caterpillars with their head capsule stack intact suggests that head capsule stacks make it more difficult for the bug to subjugate the caterpillar. Importantly, the attacks on the +HC caterpillars also provide evidence that the head capsule stack can act as a false target for predators (‘decoy mechanism’) and as a weapon to fend off the bug and deflect it’s rostrum (‘lance mechanism’), both extending the time taken for the bug to overcome the caterpillar. Prolonging the duration of the attack and subjugation phases of predation (sensu Endler, 1991) may increase opportunity for escape or the likelihood that a predator will give up on the attack. Even though all attacks by the bug were eventually successful, this should not be interpreted as indicating that head capsules do not influence attack success. The experimental setup likely restricted the effectiveness of caterpillar responses, since their confinement meant they could not escape by dropping or walking away. Such responses to attack are common in U. lugens (Low, McArthur & Hochuli, 2014) and are known to be effective means of escaping predators (Castellanos et al., 2011). Nevertheless, further predator choice trials testing a greater range of natural enemies would be useful to confirm our preliminary conclusions regarding the mechanisms by which head capsule stacking functions as a defence.

Our study suggests that head capsule stacking may be a very cost-effective way of deterring natural enemies, given that no additional biosynthesis is required beyond what would ordinarily occur in the animal. If there are indeed negligible costs to the retention of moulted head capsules, this begs the question why the behaviour is not seen more often among caterpillars. It is possible that there may be ecological or physiological costs if larvae with head capsules are more conspicuous to certain predators or if there are energetic costs of carrying the head capsules. Future work should investigate potential costs and consider the phylogenetic and ecological correlates of the trait to help to explain its occurrence and why it is not more widely observed among caterpillars.

Supplemental Information

Data S1 Raw Data

Click here for additional data file.

Figure S1 Evidence of predation

Examples of evidence of predation, likely from spiders, on caterpillars during the field experiment testing the influence of stacked head capsules on rates of predation.

Click here for additional data file.

Video S1 Predatory bug attacking Uraba lugens caterpillar

A video of an attack by the pentatomid bug on a Uraba lugens caterpillar, showing how it uses its head capsule stack to defend itself.

Click here for additional data file.

Thanks to M Low, D Attard, R Reid and A Zhuo for help in the field, G Cassis for identification of the pentatomid bug and D Britton for identification of the parasitoids. Thanks also to T Hossie, M Speed and an anonymous reviewer for helpful comments on the manuscript.

Additional Information and Declarations

Competing Interests

Author Contributions

Field Study Permissions

Data Availability

The authors declare there are no competing interests.

Petah A. Low conceived and designed the experiments, performed the experiments, analyzed the data, contributed reagents/materials/analysis tools, wrote the paper, prepared figures and/or tables, reviewed drafts of the paper.

Clare McArthur and Dieter F. Hochuli designed the experiments, contributed reagents/materials/analysis tools, reviewed drafts of the paper.

The following information was supplied relating to field study approvals (i.e., approving body and any reference numbers):

Field work was conducted under National Parks and Wildlife Services N.S.W. Scientific Licence number SL100838.

The following information was supplied regarding data availability:

Data can be found in the Supplemental Information.

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
