# Peer review of "Head capsule stacking by caterpillars: morphology complements behaviour to provide a novel defence"

_PeerJ, doi:10.7717/peerj.1714_

## Round 0.1 · original submission · Major Revisions

Dear Dr Low

Thank you for your submission, which has now been seen by three reviewers. As you will see, two were very positive and one was more critical. One of the very positive reviewers did, however, suggest an alternative analysis might be more appropriate for your data (and very kndly provided code to help you run such analysis), and the main thrust of the negative reviewer's argument is that your experiement was perhaps not as well designed or robust as it could have been, and therefore perhaps not as conclusive as you suggest. Based on my reading of the reviews and your manuscript, I can see both points of view have merit, but I don't think a study need be definitive on an issue to be published, but perhaps you need to make the preliminary nature of your results clearer. Given this, I'm afraid I can't accept your paper for publication at present, but I would be willing to consider a revised version that took our reviewers' comments into account.

with very best wishes,
Louise

·

Basic reporting

This is a very well written, and includes the appropriate level of detail. As such, the manuscript fulfills all the Basic Reporting criteria set out by PeerJ.

Experimental design

The experiments are well designed and implemented. As such, the manuscript fulfills the Experimental Design criteria set out by PeerJ.

Validity of the findings

Although the analyses are appropriate, the authors could recover more statistical power by employing a Cox proportional hazards approach. I make specific suggestions on how to do this below, and strongly encourage the authors to reanalyze their data using this technique.

Major comment: Analysis of survival data

I strongly encourage the authors to employ a different form of survival analysis called Cox proportional hazards (CPH) to reanalyze their data. The authors may not have been aware that CPH can deal with the grouped nature of their design through what is called “clustering”, given that they employed GLMs with a Poisson distribution to compare with their Kaplan-Meier analysis. CPH can also deal with another problem the authors identify on lines 239-241, that predation is highly variable across groups. For example, in the "separate batches" analysis it might be appropriate to consider that replicates 1+ and 1- would have a similar baseline hazard, meaning that the analysis should be “stratified” by replicate. Baseline hazard might also differ among host plant species, so the analysis might benefit by stratifying by host plant. Survival analysis is more powerful (i.e., lower type II errors) than a GLM, clustering deals with potential lack of independence of survival within a group, and stratifying can remove statistical noise unrelated to treatments (e.g., related to differences in predation pressure between sites).

I see in the methods that the authors used JMP (line 141). I realize that the authors may not be familiar with R and so I provide some code for the authors to use if they decide to attempt the CPH approach I suggest. John Fox’s appendix called “Cox Proportional-Hazards Regression for Survival Data” is a good primer for conducting and interpreting CPH models using R. It is freely available online here: https://socserv.socsci.mcmaster.ca/jfox/Books/Companion-1E/appendix-cox-regression.pdf

Sample R code:

########################################
library(splines) #required for survival package
library(survival) #required for Cox proportional hazards

FieldData <- read.csv("c:/R Course/field_data.csv")
#read in the data set

attach(FieldData)
#attaching data makes it easier to call variables for analyses below

model1 <- coxph(Surv(Days_alive, Killed) ~ HeadCapsule +strata(ReplicateID) + strata(HostPlant) + cluster(Group), data=FieldData)

summary(model1) # provides model output

cox.zph(model1)
#tests the assumption of proportionality. No variables should be significant.

##################################

Minor comments:

Line 221: Consider revising to “…to be effective against the full suite of natural enemies in natural systems…”

Line 241-243: I’m not convinced that the data supports this inference. Particularly the statement that head capsules are most effective under intermediate levels of predation seems problematic to establish, and not overly meaningful. Perhaps it is more reasonable to say that, while providing some level of protection, head capsules may not be equally effective against all predators, and are not sufficient to prevent predation by highly motivated predators.

Lines 255-259: Another supporting argument that the authors might want to include here or in the methods is that there is no reason to expect that there would be a systematic bias in rate of dispersal or disappearance between the two treatments.

Line 264-266: Maybe the spiders needed >24 hrs before you would observe predation? Spiders might may need to set up webbing or establish hunting areas.

Lines 269-271: Is it possible that the bugs “know” that prey with head capsules takes longer or is less likely to result in successful predation, and are therefore less profitable? Indeed a growing body of work has illustrated that predatory insects appear capable of associative learning (e.g., Guillette et al 2009 Behavioural Processes 80: 224–232).

Line 277: “sensu” should be italicized

Line 290: I think “negligible” is more appropriate than “no”

Additional comments

In this manuscript the authors examine the function of stacked head capsules as an anti-predator defence in caterpillars. Using experimental manipulations in the lab and field. They find that head capsules likely do provide a protective advantage relative to caterpillars lacking stacked head capsules, both from predators and parasitoids. Field and lab work is well grounded in detailed preliminary observational work. Analysis of field experiments were conducted in two ways, one where authors compared the survival of groups of caterpillars where the entire group possessed or lacked head capsules (ore were mixed), and another where the authors tracked the survival of groups of caterpillars composed of prey with and without head capsules. Only in the later experiment did the authors observe a protective advantage conferred by head capsules. Using staged predator-prey interactions they are able to demonstrate some support for three proposed mechanisms that could be operating: i) “illusion” where head capsules deter attack, ii) “decoy” where head capsuled direct strikes away from the true head, and iii) “lance” where the stack of head capsules used as a defensive weapon to fend off predators.

This is a very well written, the experiments are well designed and implemented, and the analyses are appropriate. As such, the manuscript fulfills all the Basic Reporting and Experimental Design criteria set out by PeerJ. Generally speaking the discussion and inferences therein are reasonably tempered, though I do have some thoughts (detailed above) that I would encourage authors to consider using to revise their manuscript. I also think the authors should consider reanalyzing their data employing a Cox proportional hazards regression approach which will recover more statistical power in their field studies than the approached currently used.

Above I provide details (incl. R code) on how to do the analysis I suggest. I have also generated an Excel file to illustrate how to structure the data files for these analyses, that I hope I can attach to this review.

Reviewer 2 ·

Basic reporting

This manuscript is well-written and the presentation is generally clear. Also, sufficient background information has been presented, although some parts of it (anti-predation adaptations) seems mainly be based on couple of decades old literature. The structure of the manuscript conforms to the standard of the journal.

Experimental design

The topic of the manuscript is interesting and clearly within the scope of the journal. Research questions are meaningful and methods have been described well. One part of the study, the predation experiment conducted in the lab, suffers from extremely low number of replication. This part could have been designed and conducted better, i.e., using a more systematic and properly replicated approach to investigate the importance of one or a few chosen, potential predator taxa. This issue is also partly related to my main concerns. The aims of the study were (1) to establish whether the stack of head capsules has a protective function and (2) to investigate three alternative hypotheses that could account for such protective effect. It has not been clearly explained how the treatment groups of the field experiment that attempted to establish whether the stack of head capsules has a protective function would serve that purpose or why the three treatments were chosen. Specifically, it is not clear to me what was the question that the treatment "Mixed" was supposed to answer. Further, the methods have not been designed well for the second aim, that is, the identification of the mechanism yielding a protective effect. There are no specific experimental treatments that would help to pinpoint the mechanism or exclude any of them, just an attempt to collect observational data in a predation experiment. As mentioned above the data provided by the experiment are not strong. Proper experiment(s) addressing the protective effect of each of the proposed protective mechanisms would be necessary to justify conclusions regarding the significance of the mechanisms.

Validity of the findings

The results are inconclusive and to some extent even contradictory considering that in one set of groups (Mixed) the individuals with head capsule stacks disappeared at a lower rate than individuals without the capsules, where comparison between treatment groups did not reveal any effect. The authors’ interpretation seems to be that the results would lend support to the protective effect of the stacks of head capsules rather than that the results would be inconclusive. Considering the significance level of the test supporting the protective effect (p = 0.048) and that the comparison between the treatment groups did not provide support for such effect, one can ask whether the data in this case are robust enough.

Additional comments

Here are some additional, detailed comments that I hope that the authors will find uselful:

Line 55: Unclear how/why clicking or whistling decreases predation risk. Or are they signals of defences? I suggest that you add a word or to two clarify the function.

L. 59: Reword. Shape disruption, color matching and counter-shading are various forms of camouflage.

L. 67: “the study of defence has largely been anecdotal and based on intuitive and subjective interpretations (Malcolm 1992; Scoble 1992)”: I agree that many older studies on anti-predator adaptations suffer from subjectivity, lack of replication etc., and unfortunately there are still some traces of such research tradition. But I also think that what you are stating here was more true 23 years ago than today. At least when considering the research on protective coloration, the field has improved and there exist a large number of rigorous, objective studies. I therefore suggest that you adjust this sentence (more encouragement, less judgement) as it currently may be interpreted that you have missed a lot of the more recent literature.

L. 74: Is it really a behaviour or a developmental phenomenon?

L. 78: “providing a false target”: For such “false target” to provide protection (i.e. to divert strikes away from the head region) suggests that the pile of head capsules is relatively large or located not very close to the head. I suggest that you include a more detailed description so that the relative size and location of the structure and its coloration in relation to the rest of the body will be clear to the reader.

L. 78: “decoy mechanism”: This type of effect is often called divertive or deflective effect.

L 118: How many of them in each container?

L. 123: Was each group placed in a different tree or were there several of them in the same tree?

L. 125-127: Explain also why you chose these three treatment groups or how they correspond to your research questions.

L. 144: Include a reference. Why is this important?

L. 176: I suggest that you present median and inter-quartile intervals because the distribution may not be normal or symmetric. Present also the number of the groups.

L. 195-198: Here it would be possible to run a statistical test to compare the dependence of the incidence of parasitism of the treatment group.

L. 201-211: Remember that the number of replicates is only 1!

L. 222: As there were no direct observations of predation, could some of the disappearances of the larvae have alternative explanations, e.g. that they (actively to avoid predation?) dropped off the leaf?

L. 223: You should provide evidence, i.e. a statistical test to demonstrate such reduction in the level of parasitism.

L. 260: I find it surprising that the manuscript does not comment on the importance of vertebrate predators (birds), particularly when some of the possible protective functions of the stack of head capsules involve predator vision.

L. 269-289: This is quite extensive discussion on this part of the study, considering that the number of replicates is one! That justifies only very limited conclusions at most. Remember also how your data compare to what you wrote on line 69-70.

L. 269-276: I think it is too much to say that this test would provide evidence for any of the three mechanisms. Possibly, it does not present evidence AGAINST any of them.

L. 270: Considering that the single predator received multiple presentations of prey, the observed choice could also be explained by learning.

·

Basic reporting

The ms is well written and clear. I have no issues here,.

Experimental design

I really like that the authors have taken their experiments into the field - this is done rarely but is invaluable in predation studies. Here the authors manipulate head stacks on caterpillars and then put them out in the field, measuring survival some time later. The experiments are clearly explained, and in my view, well designed.

The interesting result comes in the mixed group - as the authors suggest, the advantage of the defence may be seen when there are "easier" alternatives available. In effect the increased handling time from stacked individuals makes them suboptimal, if there are unstacked individuals available.

I like also that this suggests a social dimension to the defence - increasing risk for some group members, decreasing it for others. Are these animals related in groups? Probably - from the same clutches? I wonder about polyandry in the evolution of kin selected defences.

There are follow up experiments in the lab. These show that head-stacking does increase the period in which the prey can fend off the predator.

I very much enjoyed reading this section.

Validity of the findings

Yes - conclusions follow well; disucssion is thoughtful and interesting.

Additional comments

I enjoyed reading this ms.

---

## Round 0.2 · accepted · Accept

Dear Petah,

Thanks very much for your speedy revision. I think you've done an excellent job, and I'm very happy to accept your paper for publication. Many congratulations on a fine piece of work!

All the very best,

Louise